# Determinants of Pain Intensity in Physical Education Teachers Focusing on Dance Teachers: A Cross-Sectional Study

**DOI:** 10.3390/ijerph18042193

**Published:** 2021-02-23

**Authors:** Eileen M. Wanke, Jasmin Haenel, Thomas Schoettker-Koeniger, David A. Groneberg

**Affiliations:** 1Institute of Occupational Medicine, Social Medicine and Environmental Medicine, Goethe-University, Theodor-Stern-Kai 7, 60590 Frankfurt am Main, Germany; j.haenel@med.uni-frankfurt.de (J.H.); groneberg@med.uni-frankfurt.de (D.A.G.); 2Faculty of Social Work and Health, HAWK-University of Applied Sciences and Arts, Goschentor 1, 31134 Hildesheim, Germany; thomas.schoettker-koeniger@hawk.de

**Keywords:** dance teacher, musculoskeletal system, pain intensity

## Abstract

(1) Background: Dance teachers (DT) are dependent on their functional body. Pain can hardly be avoided during the professional practice of dance. Pain can become so intense that it impairs, or even prevents, the professional practice. The aim of this study was to identify the determinants of pain intensity of the most severely affected body regions of DT in pain during the three-month period prior to the survey. (2) Methods: This cross-sectional study was conducted by an online survey. A total of 166 DT participated in the study; 143 of the DT were in pain during the three-month period and were included in the analysis. Using multiple linear regression, the determinants of pain intensity were identified from population parameters, occupational data, pain localisation, and temporal pain course. (3) Results: Regions of the lower extremity and head/trunk regions were most frequently indicated as the body regions with the most severe pain. The multiple regression model generated with the factors “functional impairment”, “biomechanical exposure”, and “pain at rest” explains a statistically significant, moderate proportion of the variance in pain intensity (R^2^ = 0.22, F (3, 106) = 10.04, *p* < 0.001). (4) Conclusions: Intensity of pain in DT seems to be related to the physical demands of professional practice.

## 1. Introduction

The practice of dance as a profession requires a functioning body as a prerequisite for unrestricted participation in professional life. This applies not only to professional dancers, but also to dance teachers (DT) [1,2,3]. The work of DT includes teaching dance to children, adolescents, and/or adults [1]. Depending on the dance style to be taught, as well as the age and dancing level of the students, the physical and mental demands on DT vary [1,2,4]. For instance, teaching beginners is perceived as physically and mentally more demanding than teaching professional dancers [4]. In addition, the physical demands for the DT can vary depending on the type of dance teaching required, which can be done while seated, by indicating, or through full movement demonstration.

The high physical strain of the musculoskeletal system in dance can lead to complaints, like injuries and pain, which are provoked in particular by over- and/or incorrect strains [5,6,7]. Musculoskeletal pain is common in dance. Among professional dancers, very high prevalences of musculoskeletal pain of slightly more than 90% occur within one year [8,9,10]. Female DT also show a high 12-month prevalence of pain of 76% during and 80% within 24 h after professional practice [11]. In dance culture, pain is regarded as “normal” or “everyday”, which is why it can easily be underestimated as a warning signal [12,13]. DT often continue their professional practice even under pain [14].

Pain is a subjective perception [15]. Pain is generally described as an unpleasant sensory sensation that occurs in the context of an actual or potential tissue injury [15,16]. Acute pain is more of a warning signal of tissue injury, while chronic pain is pathological and persistent beyond the time of tissue healing [16,17]. Sensory pain intensity correlates with noxious stimulus in acute pain [15]. The level of suffering from pain (affective component) is interrelated to perceived pain intensity [15]. Pain intensity can, thus, be assumed to be an important factor influencing suffering and health. This is also reflected in the socio-culturally developed constructs of “positively” experienced performance pain and “negatively” perceived injury pain in dance. Injury pain tends to be perceived as more intense than performance pain. In the study by Anderson and Hanrahan [18], dancers rated a pain perceived as performance pain in intensity on the Numeric Rating Scale (NRS) as 5.4 (SD = 2.2) on average, while an injury pain was classified as 6.3 (SD = 2.0) on average (*p* = 0.19). In the professional practice of dance, pain becomes particularly problematic if it restricts or persists in the practice of the profession. These characteristics are attributed to “bad” pain [13,19].

How pain is perceived is determined by various situational, sociocultural, and psychological factors [15,16]. Some studies investigated associated factors of pain in professional dancers; Jacobs et al. [20] identified the years of professional dance practice among ballet dancers as a determinant of function-limiting pain at the self-estimated functional inability because of pain (SEFIP) with a score of ≥3. Furthermore, in the study by Ramel and Moritz [9], muscular tension before performances (Prevalence Odds Ratio (POR) = 4.2, 95% CI: 1.1–16.0, *p* = 0.03), as well as job satisfaction (POR = 3.7, 95% CI: 1.1–12.7, *p* = 0.03), were found to be significant influencing factors with regard to pain conditions associated with the incapacity to work. In professional ballet dancers, Dore and Guerra [21] found significant positive relationships between pain intensity on the NRS in the lower back and dancing (r = 0.23, *p* = 0.005), but also with sleep (r = 0.32, *p* = 0.001), mood (r = 0.19, *p* = 0.02), and personal relationships (r = 0.18, *p* = 0.02).

In the professional group of DT, the phenomenon of pain is still largely unexplored. There are a few descriptive studies available on musculoskeletal complaints, such as pain, from which it becomes clear that such complaints in DT are a problem that should not be underestimated [1,11,14]. Furthermore, the review by Swain et al. [22] on low back pain (LBP) and injuries in dance indicates a lack of studies that address factors influencing LBP in multivariate analyses. To our knowledge, there are no studies on multivariate analyses of DT, nor on pain outcomes other than pain prevalence, like pain intensity. A connection between the intensity of pain and the ability to work in dance can be assumed. In order to prevent an inability to work when pain occurs, or to restore the ability to work, it is very important to know about determinants that can be associated with pain intensity in DT. The aim of this study was to identify the determinants of pain intensity in the most affected region in those DT suffering from pain within a three-month period prior to the survey.

## 2. Materials and Methods

### 2.1. Study Design and Population

The data of this quantitative, retrospective observational study were collected anonymously in cross-section by means of an online questionnaire. The following inclusion criteria were defined for participation as DT:full-time or part-time dance mediating activity in the semi-professional or non-professional area, andage: at least 18 years old.

DT working in professional areas were not included in this study.

A positive ethics vote was issued by the Ethics Commission of the Goethe University Frankfurt (reference number: 25/19). An accompanying letter informed potential study participants about the purpose and content of the survey and about the persons and institutions responsible for research. The written consent was given by the participants by clicking on the “I agree” option in the consent form before beginning of the online survey. The chosen design assured voluntariness and anonymity.

### 2.2. Content of the Questionnaire

The pain aspects of the questionnaire were based on the Birbaumer and Schmidt model of pain perception and behaviour [16]. In addition, the temporal course of pain, with regard to acute and chronic pain, was taken into account according to basic literature [15,16,17]. The pain aspects were operationalised by modifying existing pain measurement instruments, taking into account dance-specific peculiarities on the basis of dance-specific literature [12,13,18,19]. The questionnaire contained question blocks with closed and partly open questions. The following blocks were considered for this data analysis:(a) pain prevalence,(b) pain localisation,(c) subjective sensation of pain, and(d) temporal course of pain.

Block (a) contained information on pain during dancing or within 24 h thereafter, taking into account the temporal occurrence of muscle ache [23], in the three-month and twelve-month periods prior to the survey (yes/no). Block (b) contained information on pain localisations by selection options of regions of the head and trunk, the upper extremity (right/left), and the lower extremity (right/left), including the most severely affected body region, radiating pain (yes/no) as well as accompanying symptoms on an ordinal 4-point Likert scale from “not” (=0) to “very” (=3) (e.g., “Restricted in mobility” and “Less resilient”). The content from block (c) relevant to the research objective of this analysis is the pain intensity on the Numeric Rating Scale (NRS) [24] determined for the most severely affected body region of the last three months. In block (d), based on the time divisions in 5 categories from “maximum 1 week” to “>6 months” according to Waddell [17], aspects of pain frequency from “once” (=0) to “always” (=3) were requested. In addition, the temporary dynamics of pain persistence from “seconds” (=1) to “>7 days” (=6) based on the German pain questionnaire [25] and the occurrence of pain as “sudden”, “creeping/over time”, “within 24 h after dancing”, or “different” [26] were surveyed. Furthermore, data on the occurrence of pain when weight-bearing, during movement, and at rest were given on an ordinal 4-point Likert scale from “never” (=0) to “always” (=3).

The questionnaire also contained the blocks “pain assessment” and “pain behaviour”, which were not included in this analysis. Following the model of Birbaumer and Schmidt [16], it can be assumed that pain assessment and behaviour are rather resulting factors from the pain components, including pain intensity as part of the sensory pain component, even if the factors are closely interrelated. Assuming that pain intensity determines pain assessment and behaviour rather than vice versa, the blocks “pain assessment” and “pain behaviour” were not included in the analysis. Therefore, the content of these blocks is not listed here.

Only those participants who reported pain in the context of dancing during the three-month period answered the questions on the detailed pain aspects. The period is based on the validated pain sensation scale according to Geissner [27]. The questions about radiating pain, accompanying symptoms, pain sensation, and temporal course were answered by the participants in relation to their most severely affected pain region of the body.

In addition, information on dance pedagogy activity was also requested: employment relationship (part-time/full-time, freelance/self-employed/employed), dance direction, years of dance and teaching experience, lessons per week, and preparations for competitions and performances in the three-month period (yes/no). Furthermore, typical population parameters (gender, age, height, weight, injuries yes/no, diseases yes/no) were collected.

### 2.3. Data Collection

The data collection took place over a period of three months between March and May 2017. A convenience sampling was used in the sampling. The acquisition of participants was carried out through various associations (Tanzmedizin Deutschland e. V. (tamed), German Professional Association for Dance Education e.V. (DBfT), Deutscher Tanzsportverband (DTV), Royal Academy of Dance (RAD), Dachverband Tanz), via e-mails to individual dance and ballet schools and via social networks on the internet. The questionnaire was created online via the survey server SoSci Survey. The practicability and quality of the questionnaire were checked in advance by representatives of the target population in a pre-test. In the pre-test, the participants could comment on the questionnaire and, for example, note any difficulties in understanding.

### 2.4. Data Analysis

The statistical evaluation was carried out with StataCorp. 2015. *Stata Statistical Software: Release 14*. College Station, TX, USA: StataCorp LP. The descriptive statistics of the variables collected on the study population, pain prevalence, and localisation were based on frequency data as well as position and dispersion measures. Skewness and kurtosis test for normality was performed to check for normal distribution of metric variables. Median ($\tilde{x})$ and interquartile distance (I_50_) were given for non-normal distributed metric variables.

The pain intensity, measured with the NRS, was considered as a dependent variable. Determinants of pain intensity in DT suffering from pain during the three-month period were determined by multiple linear regression. Variables from population parameters, dance pedagogical activity, pain localisation, and time course were included as possible determinants (independent variables). For the analysis, variables with missing values of more than 20% were excluded (concerned: “preparation for competitions” and the accompanying symptoms of redness, swelling, and warming). First, the correlations (Pearson, Spearman, Point Biserial Correlation) of the possible determinants with the NRS were calculated. For a polytomial nominal variable, a simple linear regression was determined in the context of NRS. Variables significantly correlated with NRS (*p* < 0.05) were preselected.

In order to reduce the number of preselected determinants by the formation of sum scores for variables of the same latent construct, test theoretical principles were checked. First, a strong correlation (Spearman) between the preselected variables (r ≥ 0.5) [28] was checked. For variables with a strong correlation, the Cronbach’s alpha was calculated as a reliability criterion for internal consistency [29]. A Cronbach’s alpha greater than 0.6 was considered acceptable. Furthermore, an exploratory factor analysis (EFA) of the main components was performed. The Eigen value of the first factor should be greater than 1 and that of the second factor less than 1. The loads of the variables on the first factor should be as high as possible (>0.5) [29]. When these criteria were fulfilled, sum scores were formed.

The selection of the best regression model from the preselected determinants was made by forward and backward selection using the adjusted R^2^ and the Akaike’s information criterion (AIC) and Bayesian information criterion (BIC) values [30]. For the determinants of the identified model, the unstandardised regression coefficient and its 95% confidence interval, the standardised regression coefficient beta (effects: β < 0.2 weak, β = 0.2–0.5 moderate, β> 0.5 strong), the semi-partial correlation^2 (contribution of the individual variable to the increase of R^2^), and the *p*-value were calculated [31]. For the entire model, the significance was determined using the F-test and the proportion of declared variance using R^2^ and adjusted R^2^ (proportion of declared variance: R^2^ < 0.1 weak, R^2^ = 0.1–0.2 moderate, R^2^ > 0.3 strong) [31]. The prerequisites for applying the multiple linear regression model were examined; the check for multicollinearity was performed by the variance inflation factor (VIF), which at VIF > 10 or 1/VIF < 0.1 points to a multicollinearity problem of the determinants [31]. The normal distribution of the residuals was also checked with the skewness/kurtosis tests for normality [31]. In addition, the statistical power of the multiple regression was calculated. Furthermore, the predictive margins were presented with the 95% CI of pain intensity on the NRS for the identified determinants.

## 3. Results

### 3.1. Study Population

Overall, n = 166 DT participated in the study. A total of 86.1% (n = 143) of the DT reported to have been in pain within the three-month period. During this period, 70.5% (n = 117) felt pain during exercise and 78.3% (n = 130) within 24 h thereafter.

The n = 143 DT (n = 130 females, n = 13 males) in pain during the three-month period were median 45.0 (I_50_ = 18.0) years old. The average height was 167.7 (SD = 6.5) cm and the median weight was 60.0 (I_50_ = 11.0) kg (BMI in kg/m^2^: $\tilde{x}$ = 21.3, I_50_ = 3.1). The majority of DT stated not to suffer from diseases (n = 96, 67.1%) or injuries (n = 78, 54.6%). Of all the DT, n = 106 (74.1%) were employed full-time (including n = 92 freelance or self-employed and n = 14 employed), and 27 (18.9%) worked part-time (including n = 22 freelance or self-employed and n = 5 employed). Most of the DT taught artistic dance (n = 141, 98.6%) directions with only n = 2 (1.4%) active in dance sports. On average, the DT had danced for 32.7 (SD = 12.6) years. The teaching experience was median 18.0 (I_50_ = 16.0) years. In the median, the DT taught 15.0 (I_50_ = 10.0) hours per week. Of all subjects, n = 93 (65.03%) had prepared for performances in the three-month period.

### 3.2. The Most Severely Affected Pain Region of the Body

Table 1 shows the body regions that were indicated by DT as the most severely affected region in each case. Over half of DT (n = 88, 61.5%) specified the lower extremity regions, often including the foot regions (n = 30, 20.98%) and the knee (n = 23, 16.1%), among the most affected regions. For 35.7% (n = 51) of all subjects, the head and trunk regions, including most frequently the lower back (n = 32, 22.4%), were particularly painful. Regions of the upper extremity were selected by only 2.8% (n = 4) of the DT. The following pain data, such as the NRS, referred to the specified pain localisation.

### 3.3. Determinants of Pain Intensity

The pain intensities of n = 143 DT are shown in Figure 1. The average intensity on the NRS was 5.51 (SD = 1.81, 95% CI: 5.21–5.81).

Statistical significance was found in the correlation with the NRS for seven variables (Table 2). The variables of pain localisation as “restricted in mobility” and “less resilient” (r_s_ = 0.5, *p* < 0.001), as well as the variables “pain when weight-bearing” and “pain during movement” (r_s_ = 0.6, *p* < 0.001), correlated strongly. The sum score “functional impairment” was formed from the first two variables (Cronbach’s α = 0.64, Eigen value factor 1: 1.47, Eigen value factor 2: 0.53, factor loads: 0.86). The sum score “Biomechanical exposure” was formed from the latter variables (Cronbach’s α = 0.69, Eigen value factor 1: 1.51, Eigen value factor 2: 0.49, factor loads 0.87).

Out of all the DT, for n = 99, the five preselected determinants (functional impairment, biomechanical exposure, frequency, pain at rest, and radiating pain) were completely answered. The most suitable model included the determinants “functional impairment”, “biomechanical exposure”, and “pain at rest” (adjusted R^2^ = 0.23, AIC = 377.1, BIC = 387.5).

The three determinants of the selected model were completely answered in n = 110 DT. The model explains a significant part of the variance with R^2^ = 0.22 (adjusted R^2^ = 0.199) with F (3, 106) = 10.04, *p* < 0.001. With 22%, the determinants explain a moderate part of the variance of pain intensity on the NRS. The following regression equation resulted:(1)$$\hat{\;\mathrm{NRS}}\;=\;2.55\;+\;0.26*\mathrm{functional}\;\mathrm{impairment}\;+\;0.39*\mathrm{biomechanical}\;\mathrm{exposure}\;+\;0.49*\mathrm{pain}\;\mathrm{at}\;\mathrm{rest}.$$

There was no multicollinearity as the VIF of the three determinants was below 1.3 (mean VIF = 1.16) and the 1/VIF was greater than 0.7. The residuals were normally distributed (*p* = 0.40). The statistical power of the model is 0.998.

The estimated values of the individual determinants are shown in Table 3. All three determinants contribute to a significant increase in R^2^ (*p* < 0.05). The standardised regression coefficients indicate a moderate increase on the NRS with the change by a standardised standard deviation (SD = 1) of a determinant.

The predictive margins with 95% CI of the determinants are shown in Figure 2a–c. In the cases where neither functional impairments of mobility nor resilience (sum score = 0) existed, the predictive value of the NRS was 4.63 (95% CI: 3.71–5.55). In cases where mobility, as well as resilience, were “very” limited (sum score = 6), the predictive value of the NRS was 6.18 (95% CI: 5.597–6.76). In cases where pain never occurred under biomechanical exposure (sum score = 0), the model predicted a value of 4.17 (95% CI: 3.06–5.28) on the NRS. At a maximum sum score = 6, which indicates permanent pain during movement and exercise, the predictive value of the NRS was 6.48 (95% CI: 5.77–7.19). In cases where pain “never” occurred at rest, the predictive value of NRS was 4.98 (95% CI: 4.428–5.53), whereas it was 6.44 (95% CI: 5.78–7.11) in cases where pain occurred permanently at rest.

## 4. Discussion

For the pain intensity on the NRS, three determinants could be identified for DT in pain during the three-month period: “functional impairment” (sum score of the variables “restricted in mobility” and “less resilient”) concerning accompanying symptoms of the most severely affected pain region, “biomechanical exposure” (sum score of the variables “pain when weight-bearing” and “pain during movement”) concerning the occurrence of pain under mechanical stress, and “pain at rest.” All three determinants showed a significantly moderate, positive correlation with pain intensity. It should be noted that the 95% CI of the predictive margins at the extremes of the three determinants is larger, which is why, in these areas, the prediction of pain intensity by the regression model is less precise.

Pain at rest contributed 0.06 to the increase of R^2^ in the semipartial correlation^2 (*p* = 0.01). It is plausible that pain, which is still present at rest, can be experienced more intensely as it is more likely to be a more persistent pain that is more difficult to influence. It cannot be ruled out that these are predominantly chronic pains that are no longer associated with a noxious trigger [15,16,17], since the pain is perceived even when the DT is not under physical strain, i.e., no mechanical stimulus is present.

The high significance of a functional body in DT is demonstrated by the fact that both “functional impairment” and “biomechanical exposure” have emerged as determinants of pain intensity. DT are dependent on the resilience and mobility of their body in their professional practice, which is why pain that occurs or restricts the body is probably perceived as more intense. Ramel et al. [32] validated the self-estimated functional inability because of pain (SEFIP), which collects pain intensities in the context of work ability in 14 different body regions using a functional test battery from professional ballet dancers. The average sensitivity across all body regions of the SEFIP was 78% with a specificity of 89% [32]. It becomes clear that physical functioning, pain intensity, and the ability to work in dance are related. Pain can be triggered by high mechanical stimuli [16], thus, the pain of the DT can be provoked, or intensified, under necessary strain and movements during the dance mediating activity. The biomechanical strain, functional limitations, and the perceived intensity of pain are, therefore, to be regarded as condition factors that interact with each other. The extent to which these condition factors occur might be also probably dependent on the type of dance teaching required, i.e., while sitting, by indicating, or by full movement demonstrations. Full movement demonstrations are especially necessary for less qualified students [2]. The physical workload of teaching beginners and advanced dancers is perceived by DT to be higher than that of teaching professional dancers [4]. This study was about DT from the semi- and non-professional areas. Therefore, it can be assumed that their professional activity involves more physically demanding forms of mediation, which might be the reason why complaints caused by and during movement and strain are experienced as more intense. A change of two points on the NRS can be regarded as clinically relevant [24], and especially between the extremes of the sum score of “biomechanical exposure”, clinically relevant changes on the NRS can be observed in the predictive margins.

When interpreting the results, it is important to take into account the particularities of the pain perception of dancers. The studies by Tajet-Foxell and Rose [33] and Paparizos et al. [34] showed changes in pain perception in professional and non-professional ballet dancers with an experimentally generated pain stimulus (Cold Pressor Test). In either survey, dancers showed significantly higher pain tolerances than non-dancers [33,34]. Regarding the perception of pain intensity, Paparizos et al. [34] found no significant difference between non-professional dancers and non-dancers. In contrast, the professional dancers of Tajet-Foxell and Rose [33] even perceived the pain as more intense than the non-dancers. In immediate practice, however, it should be noted that pain perception is influenced by various biospsychosocial factors [15,16]. Thus, when teaching dance, for example, further sensomotoric and cognitive processing processes can be assumed, which could affect pain perceptions. The model explains a moderate 22% variance in pain intensity on the NRS with 78% of the variance not clarified. Therefore, there are further factors to be assumed in the explanation of the pain intensities in DT that can be associated with the occupation or that can also lie outside the occupation. Various individual and situational factors, such as job satisfaction [9] or sleep, mood, and personal relationships [21], seem to influence the severity of pain and the associated possible consequences, such as the inability to work in dance. Last, but not least, pain intensity might be influenced by pain treatment, e.g., pain medication, as well as by pain management strategies in the course of pain behaviour, which was not part of this analysis.

### Limitations and Recommendations for Future Research

With regard to the questionnaire used, it should be noted that although it is based on existing theories and assessments of pain and has been pre-tested, its psychometric properties have not yet been fully validated. Initial calculations have been made in this study in order to form sum scores for variables of the same latent construct. The representativeness of the sample may be limited by the convenience sampling, as this generated a self-selective, casual smear sample [35]. A selection bias is possible. Due to the cross-sectional design, the results of this analysis cannot be interpreted in the course of a cause-and-effect relationship. However, it can be assumed that the pain intensity is interrelated with the identified determinants rather than cause-effect related. Furthermore, a recall bias may be present from the self-reported pain data over the three-month period prior to the survey. Information on the workload of the DT only included the weekly lessons. In addition, information, such as the age and dancing level of the students and the primary type of dance teaching (sitting, indicating, full movement demonstration), was not part of the data analysis. Furthermore, pain behaviour, e.g., the treatment of pain, was not included in the analysis.

Future studies should aim at random sampling. Longitudinal studies are recommended to generate knowledge about the causality of factors. Furthermore, information on the age and dancing level of the students, on the type of dance teaching and pain behaviour should be included. These factors may explain aspects of the variance in pain intensity among DT. A qualitative, hypothesis-generating study design with the research approach of Grounded Theory methodology can provide a promising approach to gain further knowledge. Many studies are primarily based on pain influencing factors associated with professional dance activities (job satisfaction, etc.). In so far as both occupational and non-occupational factors can influence the pain perception of DT, further factors and the interaction of factors can be elucidated via a Grounded Theory.

## 5. Conclusions

The perceived pain intensities of the most severely affected body regions of DT in pain during the three-month period are essentially associated with the occurrence of pain at rest and under biomechanical exposure as well as with the perception of functional restrictions in mobility and resilience. The latter two factors, in particular, point to the importance of a functional body in the professional practice of DT. Physical stress factors, such as weight bearing, physical limitations in mobility, and resilience, seem to correlate closely with pain intensity.

A practical perspective that can be drawn from the results of this study is that when pain occurs in DT, a rehabilitative approach must be taken to restore and/or optimize the mobility and resilience required for the job.

## Figures and Tables

**Figure 1 ijerph-18-02193-f001:**
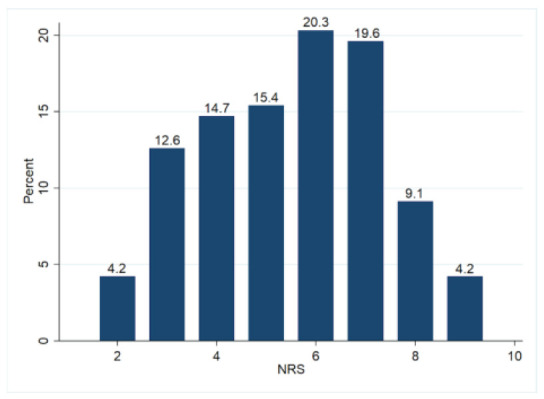
Frequency of pain intensities on the Numeric Rating Scale (0 = no pain; 10 = worst conceivable pain) in the most affected body regions of dance teachers (n = 143).

**Figure 2 ijerph-18-02193-f002:**
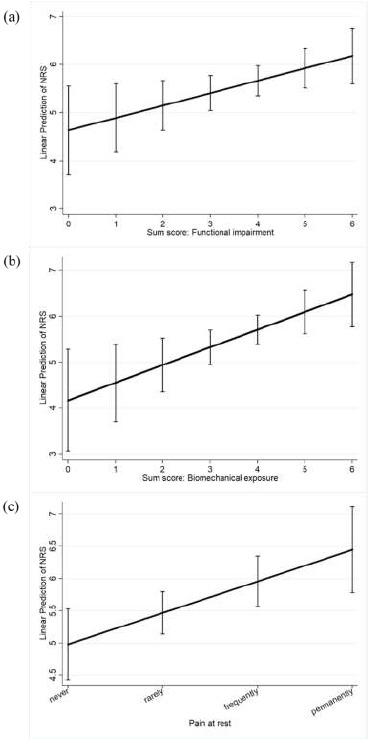
(**a**–**c**). Predictive margins with 95% CIs of pain intensities on the NRS (0 = no pain; 10 = worst conceivable pain) for the determinants (**a**) functional impairment, (**b**) biomechanical exposure, and (**c**) pain at rest.

**Table 1 ijerph-18-02193-t001:** Most severely affected pain regions of dance teachers in pain during the three month period (n = 143).

Body Region	n (%)	
**Head and trunk** (more generalised)	3 (2.1)	
Head	0	
Neck/cervical spine	8 (5.6)	
Upper back/thoracic spine	5 (3.5)	
Breast	0	
Lower back/lumbar spine/iliosacral joint	32 (22.4)	
Stomach	0	
Buttocks/pelvis	3 (2.1)	
**Upper extremity** (more generalised)	0	
	**Right**	**Left**
Shoulder	1 (0.7)	2 (1.4)
Upper arm	0	0
Elbow	0	0
Forearm	0	0
Wrist	0	1 (0.7)
Hand	0	0
**Lower extremity** (more generalised)	6 (4.2)	
	**Right**	**Left**
Hip joints	10 (6.99)	5 (3.5)
Groin	1 (0.7)	5 (3.5)
Anterior upper thigh	1 (0.7)	0
Posterior upper thigh	0	1 (0.7)
Knee	12 (8.4)	11 (7.7)
Anterior lower thigh/shinbone	0	0
Posterior lower thigh/calf	4 (2.8)	2 (1.4)
Ankle joint	4 (2.8)	5 (3.5)
Back-/midfoot	7 (4.9)	6 (4.2)
Forefoot and toes (except big toe)	3 (2.1)	2 (1.4)
Big toe	2 (1.4)	1 (0.7)

**Table 2 ijerph-18-02193-t002:** Correlations of independent variables with pain intensity on the NRS in dance teachers (n = 143).

Variables	Correlation Coefficient	Missing n (%)	*p*-Value
**Population parameters**			
Age	r = −0.05 ^a^	-	0.54
Body mass index	r = −0.001 ^a^	4 (2.8)	0.99
Gender	r_pbis_ = −0.14 ^c^	-	0.09
Diseases	r_pbis_ = 0.15 ^c^	1 (0.7)	0.07
Injuries	r_pbis_ = 0.09 ^c^	1 (0.7)	0.31
**Information on dance teaching**			
Years of dancing	r = −0.03 ^a^	4 (2.8)	0.74
Years of dance teaching	r = −0.004 ^a^	2 (1.4)	0.96
Hours per week	r = 0.07 ^a^	4 (2.8)	0.39
**Pain localisation**			
Radiating pain *	r_pbis_ = 0.19 ^c^	-	0.03
Tight/hard/tense	r_s_ = 0.13 ^b^	20 (14.0)	0.17
Restricted in mobility * Less resilient * Sum score: Functional impairment *	r_s_ = 0.34 ^b^ r_s_ = 0.35 ^b^ r = 0.37 ^a^	9 (6.3) 7 (4.9) 15 (10.5)	0.00 0.00 0.00
**Temporal course of pain**			
Duration	r_s_ = 0.07 ^b^	14 (9.8)	0.46
Frequency *	r_s_ = 0.28 ^b^	14 (9.8)	0.00
Pain persistence	r_s_ = 0.07 ^b^	2 (1.4)	0.43
Pain occurrence	r = 0.05	-	0.94 ^d^
Pain when weight-bearing * Pain during movement * Sum score: Biomechanical exposure *	r_s_ = 0.36 ^b^ r_s_ = 0.26 ^b^ r = 0.34 ^a^	16 (11.2) 7 (4.9) 19 (13.3)	0.00 0.00 0.00
Pain at rest *	r_s_ = 0.23 ^b^	13 (9.1)	0.01

^a^ Pearson (r), ^b^ Spearman (r_s_), ^c^ Point Biserial Correlation (r_pbis_), ^d^ simple linear regression (r = √R^2^), * *p* < 0.05.

**Table 3 ijerph-18-02193-t003:** Estimates of determinants of multiple regression of pain intensity on the NRS in dance teachers (n = 110).

Determinants	Regression Coefficient		Semipartial Correlation^2	*p*-Value
	Unstandardised [95% CI]	Standardised (beta)		
Functional impairment	0.26 [0.31–0.48]	0.21	0.04	0.03
Biomechanical exposure	0.39 [0.10–0.67]	0.25	0.05	0.01
Pain at rest	0.49 [0.14–0.84]	0.24	0.06	0.01

## Data Availability

The datasets used and analysed during this study are available from the corresponding author on reasonable request.

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
