# Peer review of "Determinants of Pain Intensity in Physical Education Teachers Focusing on Dance Teachers: A Cross-Sectional Study"

_ijerph, 2021, doi:10.3390/ijerph18042193_

Round 1

Reviewer 1 Report

Presented study aimed to identify the determinants of pain intensity of the most severely affected body regions of dance teachers in pain during the three month period prior to the survey.

The results highlighted that regions of the lower extremity and head/trunk regions were most frequently indicated as the body regions with the most severe pain.

All factors are described clearly. However, I have several suggestions:

  1. Material and Methods (2.2. Sample calculation and randomization)

The following inclusion criteria were defined for participation as DT:

  • full-time or part-time dance mediating activity in the semi-professional or non-professional area, and age: at least 18 years old.

Please state if any exclusion criteria were applied when selecting the group.

  1. 2.2. Content of the questionnaire: 141-142

The information on radiating pain, accompanying symptoms, pain sensation, and temporal course related to the most severely affected pain region of the body.

Please revise

  1. 3.2. The most severely affected pain region of the body
    Please try to summarize the data in the table with all body sites included in the study. This information may be relevant.

Overall:
In general, the work is very interesting and can make a significant contribution to literature.

I hope my suggestions will help improve this paper.

Author Response

Reviewer 1

Open Review

English language and style

( ) Extensive editing of English language and style required
( ) Moderate English changes required
(x) English language and style are fine/minor spell check required
( ) I don't feel qualified to judge about the English language and style

Yes

Can be improved

Must be improved

Not applicable

Does the introduction provide sufficient background and include all relevant references?

(x)

( )

( )

( )

Is the research design appropriate?

(x)

( )

( )

( )

Are the methods adequately described?

( )

(x)

( )

( )

Are the results clearly presented?

( )

(x)

( )

( )

Are the conclusions supported by the results?

(x)

( )

( )

( )

Comments and Suggestions for Authors

Presented study aimed to identify the determinants of pain intensity of the most severely affected body regions of dance teachers in pain during the three month period prior to the survey.

The results highlighted that regions of the lower extremity and head/trunk regions were most frequently indicated as the body regions with the most severe pain.

All factors are described clearly. However, I have several suggestions:

  1. Material and Methods (2.2. Sample calculation and randomization)

The following inclusion criteria were defined for participation as DT:

full-time or part-time dance mediating activity in the semi-professional or non-professional area, and age: at least 18 years old.

Please state if any exclusion criteria were applied when selecting the group.

  • We have now also specified the exclusion criteria. Thank you.

  1. 2. Content of the questionnaire: 141-142

The information on radiating pain, accompanying symptoms, pain sensation, and temporal course related to the most severely affected pain region of the body.

Please revise

  • The participants first indicated their most severely affected pain region within the last three months. They were informed that all subsequent questions about the pain refer to the indicated region. We have now rephrased the sentence in the manuscript and hope that this is clearer now.
  1. 2. The most severely affected pain region of the body
    Please try to summarize the data in the table with all body sites included in the study. This information may be relevant.
  • Thank you for this good suggestion. We have added a table.

Overall:
In general, the work is very interesting and can make a significant contribution to literature.

I hope my suggestions will help improve this paper.

  • Thank you very much for your valuable suggestions. We have applied them to the methods and results section.

Submission Date

30 January 2021

Date of this review

08 Feb 2021 21:56:52

Reviewer 2 Report

The article is well written and the topic is interesting. Anyway the statistical analysis was not properly performed. The authors reported this in the limitations. They recommended a longitunal analysis, as they should have done but the authors didn't. Why? Moreover, none of the methods they used took into account repeated observations within dance teachers i.e. in the three-month and twelve-month periods prior to the survey, 

I will suggest a more adeguate analysis, taking also into account other factors /confounding correlated with NRS.

Moreover, the authors also should have described or discussed how the dance style influenced the results, even if the majority (how many? not reported) taught artistic dance.

In addition, the sampling was nonprobabilistic. Could you better describe this? Were there a sort of "inclusion criteria"? Was a possible selection bias taken into account?

Finally, the visual aspect of Figures should be improved. They look like first draft of Figures as internal use rather than Figures for a publication. Colors in panel, for example, should be removed.

Author Response

Reviewer 2

Open Review

English language and style

( ) Extensive editing of English language and style required
( ) Moderate English changes required
(x) English language and style are fine/minor spell check required
( ) I don't feel qualified to judge about the English language and style

Yes

Can be improved

Must be improved

Not applicable

Does the introduction provide sufficient background and include all relevant references?

(x)

( )

( )

( )

Is the research design appropriate?

( )

( )

(x)

( )

Are the methods adequately described?

( )

( )

(x)

( )

Are the results clearly presented?

( )

(x)

( )

( )

Are the conclusions supported by the results?

(x)

( )

( )

( )

Comments and Suggestions for Authors

The article is well written and the topic is interesting. Anyway the statistical analysis was not properly performed. The authors reported this in the limitations. They recommended a longitunal analysis, as they should have done but the authors didn't. Why? Moreover, none of the methods they used took into account repeated observations within dance teachers i.e. in the three-month and twelve-month periods prior to the survey, 

  • Thank you very much for your comments. The study includes initial baseline data on pain among dance teachers. After extensive research, we found it most useful to conduct a multiple linear regression in relation to our research question. The regression model is a first idea of which determinants might be related to pain intensity among dance teachers. Of course, these need to be further validated and explored. You are right that there are limitations due to the design, such as the fact that the determinants cannot be interpreted in sense of a cause-effect relationship. As you have already noted, we have addressed these limitations in the paper. In order to make repeated observations, a new study would be necessary with a longitudinal design. We have pointed out the necessity of conducting such a study in our recommendations for future research.

I will suggest a more adeguate analysis, taking also into account other factors /confounding correlated with NRS.

  • We correlated all variables, that could be related to the NRS, with the NRS in the course of pre-selecting the determinants (see Table 2; in the original manuscript version Table 1). Most of the variables did not show a significant correlation with the NRS.
  • We were able to determine 22% of the variance based on the model. We have referred to possible factors that should be collected in further studies in our recommendations for future studies. We have now added a sentence that refers to these further factors in the determination of the variance in the NRS. Thank you.

Moreover, the authors also should have described or discussed how the dance style influenced the results, even if the majority (how many? not reported) taught artistic dance.

  • We have now added the frequency data for dance teachers who teach artistic dance classes. Since the proportion of those who are active in dance sport (n=2) is very small (individual cases), the dance style in the sample can be considered homogeneous.

In addition, the sampling was nonprobabilistic. Could you better describe this? Were there a sort of "inclusion criteria"? Was a possible selection bias taken into account?

  • Thank you for the hint. A probably more common word is "convenience sampling ". We have renamed this in the paper.
  • We have now also highlighted the possibility of selection bias in the limitations.

Finally, the visual aspect of Figures should be improved. They look like first draft of Figures as internal use rather than Figures for a publication. Colors in panel, for example, should be removed.

  • We have revised the figures and removed the colors in panel.

Submission Date

30 January 2021

Date of this review

05 Feb 2021 21:52:03

Round 2

Reviewer 2 Report

The study has several limitations. But they are reported in the manuscript. The manuscript has been improved though not so much has been done.